# Connecting Roma Communities in COVID-19 Times: The First Roma Women Students’ Gathering Held Online

**DOI:** 10.3390/ijerph19095524

**Published:** 2022-05-02

**Authors:** Emilia Aiello, Andrea Khalfaoui, Xavier Torrens, Ramón Flecha

**Affiliations:** 1Department of Sociology, Universitat Autònoma de Barcelona, 08193 Barcelona, Spain; 2Faculty of Education and Sport, University of Deusto, 48007 Bilbao, Spain; andrea.khalfaoui@deusto.es; 3Department of Political Science, Constitutional Law and Law Philosophy, University of Barcelona, 08007 Barcelona, Spain; xaviertorrens@ub.edu; 4Department of Sociology, University of Barcelona, 08007 Barcelona, Spain; ramon.flecha@ub.edu

**Keywords:** Roma women, COVID-19, Roma Women Students’ Gathering, Roma Association of Women, Roma community, dialogic online spaces, solidarity networks

## Abstract

COVID-19 has exacerbated the vulnerability of the Roma communities in Europe. However, these communities have a strong sense of resilience, and the role of Roma women must be highlighted since they have historically nurtured solidarity networks even in the most challenging situations. Aim: A particular action organized by a Roma Association of Women is analyzed: the Roma Women Students’ Gathering (RWSG, or gathering). In its 19th edition, this democratic space aimed at tackling the challenges the pandemic has raised and its impact on the Roma communities. Method: The 19th RWSG, which was the first one held online, was inductively analyzed to gain a deeper understanding of the key aspects that the Roma women highlight when they organize themselves. Results: RWSG generates optimal conditions where Roma women identify the challenges affecting their community and, drawing on the dialogues shared, agree on strategies to contest them. RWSG also enhanced solidarity interactions that enabled the conquering of the virtual space, transforming it into an additional space where the Roma could help each other and thus better navigate the uncertainties unleashed by COVID-19. Key features of the Roma culture emerged in these spaces of solidarity, such as protecting the elderly and prioritizing community wellbeing rather than only the individual’s preferences. Conclusions: Roma women play a key role in weaving an organized response to the uncertainty derived from COVID-19, and connecting them to the public sphere, potentially achieving social and political impacts.

## 1. Introduction

Since the COVID-19 breakout in early 2020, the EU Commissioner for Equality called on EU Member States to implement urgent measures for Roma communities, because COVID-19 exacerbates their exposure to structural inequality. However, despite these requests and the clear mandates of international and European human rights treaties guaranteeing equality, non-discrimination, and dignity for all people, there has been a frightening escalation of populism and racism in Europe [1]. In this regard, the Roma population suffers from the toughest discrimination since it is a social group that experiences higher rates of racism in Europe [2]. Moreover, COVID-19 has worsened discrimination and anti-Gypsyism across diverse countries as, systematically, the Roma have been blamed for contributing to the spread of the virus [3].

Even before COVID-19, Roma populations across Europe experienced significantly worse health when compared with majority populations [4]. This includes lower self-reported health and higher mortality risk [5]; lower life expectancy [6]; increased burden of communicable diseases [7]; increased morbidity from non-communicable diseases [8]; and poorer infant and child health. Moreover, the Roma people attend general medical check-ups less frequently than non-Roma populations and are more likely to attend for a specific problem rather than in a preventive way [9]. Nevertheless, Roma women’s situation is worthy of special mention since they are more likely to use health services for their children rather than themselves. This is coherent with the so-called “triple discrimination” that they have been suffering for decades: for being women, for being from an ethnic minority, and for overall having a lower academic level [10]. According to the latest international data, 62% of Roma youth are not in education, employment, or training, compared to 10% of youth in the general population.

Research shows that health literacy is a major contributor to health promoting behaviors [11]. In this vein, low education and literacy levels among the Roma exacerbate their inequalities in health and health service engagement, which are set against a background of widespread disadvantage and discrimination in their day-to-day lives, such as a lack of adequate housing or restricted access to employment. As McFadden and colleagues found, Roma populations across Europe struggle to exercise their right to healthcare on account of multiple barriers related to other determinants of disadvantage, such as low literacy levels and experiences of discrimination [12]. Existing research points out that differences in the social determinants of health evidence the inequalities that the Roma experience in accessing good-quality education or the labor market, among other public spaces that have an impact on people’s health [13]. These barriers highlight the fact that Roma people still have poor health quality and low education and literacy levels in comparison to the general population. This is even after the institutional efforts and programs put in place at a national and European level to overcome this situation. As decades of research have widely demonstrated, health and education are intertwined elements in human development [14]. Linked to this, international data show that simultaneous disparities in both areas in vulnerable groups result in a trapping cycle of exclusion for Roma communities worldwide.

Particularly in a unique situation such as the COVID-19 pandemic, many vulnerable communities have sought to put in place strategies and actions that could protect everyone’s health while ensuring the best (both in education and health) for future generations. There is a clear institutional commitment regarding the EU Member States’ responsibility for acting to protect the health and education rights of all people within their jurisdictions, including their Roma populations [3]. However, even when top-down actions have been implemented, these still fall short of guaranteeing Roma women’s academic success and health improvement. As claimed by Roma women themselves, both academic research and institutional programs aimed at improving the living conditions of the Roma communities should look at the grassroots level, analyzing how the Roma communities are organizing on the ground, listening to their voices, and eventually identifying the lessons learned from these experiences and how they can be replicated in other social contexts [15].

Linked to the above-mentioned aspects, in the last decade, researchers have responded to an increasing social demand to render science more socially responsible by having a deeper impact on society at large and by establishing innovative links between science and lay people [16,17]. In this sense, evidence points out that access to education and scientific knowledge is associated with better health, the empowerment of citizens, and active participation in public affairs, fostering a more engaged citizenry and improving employability [18]. Scientific knowledge is often inaccessible to society at large but more so to the most disadvantaged groups, such as the Roma and particularly Roma women. However, recent studies reveal how specific dialogue-based settings can effectively promote scientific self-literacy [19]. This is coherent with the movement to democratize expert knowledge and widening tools that are used for improving the scientific literacy of citizens, which facilitates scientifically informed decisions that ultimately boost communities’ health and education [20].

Several strategies have been proposed for citizens to have access to the latest advances in scientific research, advancing towards an inclusive communication of science, and thus supporting people to be themselves the ones who directly engage with science [21,22,23]. In this vein, there are numerous initiatives led by both public and private organizations, such as associations, schools, museums, and non-governmental organizations, oriented to promote access to science in various attractive ways, such as organizing scientific dialogic gatherings, science clubs run by scientists and implemented in schools, virtual museums, or engaging in scientific evidence platforms, to mention a few [23]. Many of these initiatives not only provide citizens with access to knowledge but also aim to empower them through social learning [24,25].

### The Roma Association of Women Drom Kotar Mestipen and the Roma Students’ Gathering

The Roma Association of Women Drom Kotar Mestipen (which, in Romano, means “A road for freedom”) was created in 1999 in Barcelona by Roma and non-Roma women with the purpose of enhancing Roma women’s access to and participation in educational, social, and cultural spaces [9]. The Drom Kotar Mestipen (hereinafter, DKM) was the first grassroots Roma women’s organization constituted in Catalonia, and during the last two decades, it has been leading socio-educational projects at the local and regional level, but also collaborating and leading projects and initiatives with similar organizations based across Europe, promoting the voices of grassroots Roma women, and working for their educational success as a way to improve gender equality and help break the cycle of poverty.

At present, the work done by DKM has been recognized by local, national, and European authorities for its social impact achieved with Roma women and the Roma community, enhancing their access to education and the formal job market. Examples of this recognition include the award received by DKM for its project EDUCA-ROM—Inclusive teaching material for adults: the Roma (2005–2007). EDUCA-ROM was a Grundtvig project whose objective was the promotion of inclusive education for adults in Europe, in four different countries and five organizations.

Among the core activities organized by DKM, the Roma Women Students’ Gatherings (hereinafter RWSG or gathering) are a particular activity within the strategic working lines of the organization. Every year, at least once, the RWSG is organized, involving very diverse Roma women from different contexts. One key feature of the gathering is that a unique dialogic space is created, in which every voice is considered essential. Within this dialogic atmosphere, all participants can share their multiple concerns about their own and their children’s education, discussing the barriers faced within the educational system and which actions are needed to overcome them. Moreover, the intergenerational factor is at the heart of the RWSG, since, every year, Roma girls, adults, and elderly women share their experiences in education, which contributes to meaning creation for those Roma girls who are still in school and often experience the burden of exclusionary and segregationist educational systems [26]. Roma youths and adults who have experienced early school leaving can engage in fruitful dialogues with other Roma women with successful trajectories, setting up high educational expectations for all.

The gatherings are not held in the DKM headquarters. Instead, they are normally held at a school in a neighborhood where there is a large Roma community. Moreover, a small group of Roma women who want to lead the action in their territory work hand in hand with DKM, organizing the event; this local team constitutes the “local organizing committee”. This feature is key to ensure the impact of the activity at the local level, and to promote a sense of ownership of the activity. As explained by existing research, this very element promotes a sense of ownership of the activity among grassroots Roma women, and becomes a driver for boosting grassroots Roma women’s participation and the sustainability of the gathering’s impact on the ground [27].

In all, although some experiences have been reported, less is known about how Roma women organized themselves during the COVID-19 lockdown to identify their most pressing challenges and scaffold evidence-based solutions to ultimately bring to the policy sphere their demands. The DKM has been active since the very beginning of the pandemic, serving as a space where grassroots Roma women could help each other and thus be better equipped to navigate the online virtual space. Exploring how this took place is crucial to overcome the stereotyped image of Roma women as subjugated to men and passive towards social challenges affecting their communities, thus drawing pathways to strengthen the Roma community’s resilience. Features of Roma culture are at the heart of the Roma women’s movement and of Roma feminism, such as caring for the elders, a strong sense of community and cohesion, and the extended family [28]. These unique features might be replicable and transferable beyond the Roma community to the entirety of society.

## 2. Materials and Methods

The present study focuses on exploring the 19th edition of the Roma Women Students’ Gathering organized by the Drom Kotar Mestipen association, held online in July 2020. Two inter-linked research questions guided this research: first, in what ways was the RWSG organized by DKM convened as a dialogic space for the agency expression of the grassroots Roma women participating in it? Second, what are the underlying elements that allowed grassroots Roma women to successfully navigate the virtual space, to join and participate in the RWSG, and, more broadly, to help each other during the lockdown?

The study is framed within the communicative methodology of research [29]: a methodological approach characterized by its dialogic orientation that has been widely recognized for its transformative potential. This methodology is a tool to “explain, understand, and interpret social situations which aims at changing society, -driven by utopian dreams of equality and justice” [30] (p. 237). Previous research has widely demonstrated how the communicative approach contributes to the scientific, social, and political impact of research results in those research investigations done with the Roma community, such as the case of the FP5 WORKALO [31] or the FP6 INCLUD-ED research projects [32]. This is due to the strategies of the communicative approach to directly include the researched subjects throughout the process of investigation. Examples of them are counting on the participation of Roma researchers in the research teams, or designing advisory councils formed by community members with a central role at the time of deciding research strategies [33]. The inclusion of research subjects as part of the research team ensures that the voices of researched communities are considered, since doing so is one way to depart from the conception of researched subjects as mere objects of investigation. It is worth mentioning that one of the co-authors of this article is a young Roma scholar herself (Dr. Andrea Khalfaoui).

### 2.1. A Glimpse into the 19th Online Roma Women Students’ Gathering

Twenty-five Roma women aged between 9 and 67 years old took part in the 19th RWSG held online, including five who were invited to give a short talk linked with the theme of the gathering, which was “Education and health, we do not want to choose between them, we want them both!” This motto was chosen by the Roma women members of the local organizing committee in dialogue with members of the DKM association during the previous dialogues regarding their priorities and concerns about the negative impact that the lockdown imposed during March–June 2020 in Spain could have on Roma children.

Two academic women with expertise in inclusive education and who had, for a long time, worked with the Roma community were invited to join the opening session of the online gathering, sharing their expertise on those actions and strategies that could be implemented to prevent Roma children from disengaging from school and to alleviate the potential consequences of the lockdown on academic success and learning. Moreover, five Roma grassroots women from different towns were invited to share their stories regarding how they were organizing and strategizing at the local level to help others to access key information and resources related to health and education. After these interventions, the debate was opened up to anyone who wanted to share ideas, opinions, and priorities.

The director of the Program for the Roma Community and Social Innovation from the Barcelona City Council shared a message of hope linked to the power of education for everyone and especially for Roma women to achieve equality. The event was closed by a member of DKM, who also shared a message of hope, strength, and solidarity, all values that lie at the heart of the Roma cultural identity.

### 2.2. Study Procedure

The 19th RWSG lasted 2 h and 15 min, it was entirely recorded, and the DKM stored this recording safely in a hardware device. Conclusions and take-aways of the event were drafted during the gathering. For this study, DKM shared the recording of the event, as well as the conclusions and other working notes, with the research team. Moreover, two of the co-authors of this article joined the event as participants and observers. They did not intervene in the content of the event at any time, but helped by providing technical support to the event organizers when directly requested.

The whole recording of the event was watched, reviewed, and analyzed twice by the research team. In addition, the Twitter channel of the DKM association (@DromKotar) was revised to gain an understanding of what was highlighted by the online participants and other individuals interacting through social media during the time of the gathering, as well as in the previous and following weeks of the event. The news published in El Diario Feminista [34] about the RWSG was also read and analyzed by the research team for research purposes.

### 2.3. Analysis Strategy

The research team reviewed the full video and transcribed verbatim key interventions related to the underlying themes of the debate, moments that generated more agreement, as well as others that generated more acceptance during the gathering.

Then, the two main thematic dimensions that guided the analysis procedure were identified and discussed among the researchers. These were, first, the drivers for discussing the challenges derived from the lockdown with a focus on Roma children’s education (linked to research question 1, about the RWSG as a dialogic space of agency expression); and second, the potential underlying elements that allowed grassroots Roma women to successfully navigate the virtual space, and which go beyond the gathering itself to the entire Roma community (linked to research question 2). Once this was done, the interventions of the participants were clustered into these comprehensive dimensions, and further categorization was conducted to gain an accurate insight into the potential impact of the RWSG. The following broad categories were defined, and information was classified accordingly: navigating virtual space, women leadership, solidarity networks, access to COVID-19 information provided by public health authorities, key Roma cultural traits observed.

Following the communicative approach for data analysis, data were in turn analyzed considering the transformative and exclusionary dimension of the analysis [35]. As for the transformative dimension, we considered those elements shared by the women related to actions and strategies that allowed them to better navigate the uncertainties and challenges posed by the enforced lockdown at the time of helping their children in school and building alliances with other Roma women or key community agents that facilitated their access to relevant information related to health and education. As for the exclusionary dimension, we considered those elements that prevented Roma women from navigating the virtual space, as well as others related to the interpersonal and structural barriers faced by Roma children in engaging in educational activities during lockdown, as expressed by the Roma women participating in the gathering. Note that, for this research article, we mostly focused on the transformative dimension of analysis. This was because most of the already published literature on Roma people in the time of COVID-19 has tended to analyze and showcase exclusionary elements. Although acknowledging the relevance of these type of materials, our scientific and personal interest was in capturing those instances of Roma women’s human agency.

Table 1 below shows the coding scheme that guided the analysis: 

### 2.4. Ethics

To protect the participants’ identities and personal data, pseudonyms have been used. Consent forms with detailed information about the study and an explanation that they had the right to withdraw from it at any time were provided to all the persons participating in the research. The Roma Association of Women, Drom Kotar Mestipen, granted explicit consent and authorization to the co-authors to use any type of public material, such as Twitter screenshots published by the association, whenever they were used for research purposes.

The CREA Ethics Committee (Community of Researchers on Excellence for All) approved this study, Registration Nr: 20220228. In addition, all information gathered under the Narratives4Change project (EU-funded project, Project Nr 43855), in which this specific study is framed, complies with the Ethics Appraisal Procedure required by the Horizon 2020 research program, funded by the European Commission. Accordingly, the Narratives4Change project follows the Regulation (EU) 2016/679, within the EU’s new General Data Protection Regulation (GDPR).

## 3. Results

This section is organized according to the dimensions that guided the analysis. After considering and discussing the contributions of the participants, each dimension highlighted particular elements that altogether give a nuanced picture of the impact of RWSG in overcoming challenges derived from the lockdown, along with the features that scaffolded it.

In this section, we present and discuss the results of the study, focusing on how the RWSG became a space where participating Roma women identified the challenges affecting their community and, beyond this, shared strategies to cope with and contest them, showing that they can also successfully navigate the virtual space. Solidarity networks among Roma women, protecting the elderly, and prioritizing the community wellbeing, rather than just that of individuals, are common aspects underlying the themes that emerged in the gatherings. Findings reveal that the RWSG became a space for the expression of Roma women’s individual and collective agency.

### 3.1. “If We Cannot Meet in Person, We Do It Online”: On the Need for Organizing the Online RWSG

The 19th RWSG was the first time that this type of gathering was held online. When DKM asked, through phone calls, some of the women already participating in its activities whether they would like to hold the gathering online or whether this would not be worthwhile, everybody emphasized the need to hold it, even online. These women detected the need to come together as a way to open up a space of dialogue where they could discuss their thoughts, experiences, and priorities regarding the consequences of the measures implemented to stop the spread of COVID-19 during the hard lockdown imposed in Spain (from 15 March to 21 June 2020).

In this way, for DKM, it was crucial to ensure that one of the key principles of this activity, egalitarian dialogue, would also be ensured if the activity was going to be organized online. Egalitarian dialogue requires the creation of conditions that ensure that every participant can have a say at any time regarding the topic that is being discussed. It requires all opinions and interventions to be equally valued, regardless of whether it is someone with academic credentials or someone with poor literacy skills. For this to happen, DKM started organizing the event much earlier than the day the event took place, in a structured way, allocating time and scheduling meetings with members of the local organizing committee (LOC) of the gathering. In this case, meetings with the LOC were held from the end of April 2020 and were key to identifying the priorities of the Roma families at that time during the pandemic. The dialogues held in these previous meetings included talking about the difficult personal situations of each woman and their families and how the DKM could support the work that Roma families were doing to maximize its impact.

As a result of these dialogues, Roma women decided to meet weekly for a “Romi coffee” facilitated by DKM. This was an online virtual coffee space where they could continue the exchange of thoughts. It eventually became a space for supporting Roma children with their homework. This dialogic culture fostered by DKM paved the way to shaping the space of the gathering, where Roma women participants explained that what worried them the most was how to ensure that their children were following online learning when schools were closed due to COVID-19 protocols, how to navigate social services in those cases when this was needed, or how to manage the lack of economic resources exacerbated by the lockdown.

Once the date that the gathering was going to take place was chosen, the main topic that would guide the discussion of the gathering had to be agreed upon. As explained by Cristina, the president of DKM, when opening the RWSG, the motto of this unusual virtual gathering was established as “Education and health, we do not want to choose between them, we want them both!”. Women participating in the gathering explained that this was the idea that better reflected their major concern, i.e., that Roma communities felt that public authorities were sending them the message of having to choose between keeping schools open and facing the risk of being infected, or staying at home and not contracting COVID-19. Although most Roma families—as well as non-Roma families—accepted without question the idea of closing schools due to such a public safety issue, there was scant debate about how schools would ensure that vulnerable children such as the Roma could stay engaged with online learning. Dolores, a Roma woman, explained in the gathering that when this idea was indirectly communicated by public authorities, those in positions of power did not consider the negative long-lasting impact that dropping out of school could have on the lives of Roma children and their future families.

Roma people were the ones strictly following the norms of the lockdown because many of them lived with elderly relatives at home. As Ana said, they feared causing them to contract the virus. However, they also knew that if their children were not ensured a high-quality education, and if they dropped out school, it was very likely that most of them would not return to formal education. Taking this risk would have had devastating consequences for many Roma adolescents, and especially for those who were in the final stages of compulsory education.

### 3.2. The Gathering as a Space to Show How Roma Women Are Taking the Lead in Situations of Uncertainty: Building Alliances to Not Leave Anyone Behind

Evidence collected shows that participants with different profiles and coming from diverse areas worked together during the months before the RWSG, bringing their expertise and knowledge to the activity. Roma women from the most deprived neighborhoods in Catalonia worked hand in hand with professional Roma women, with those participating in different Cultos (spaces of faith), and with non-Roma women who collaborated and were members or volunteers of the DKM association.

Particularly in its 19th edition, the gathering succeeded in highlighting the role of the Roma women as the key element to mobilize and organize their communities on the ground. They also emphasized Roman women’s role at the policy level in setting the main priorities for them to overcome the consequences of the lockdown. In this vein, a non-Roma female scholar and a Roma female junior researcher joined the opening session of the gathering, sharing and discussing the latest and most important scientific evidence related to education and health that could potentially respond to the motto of the event.

After the opening session, the roundtable of experiences took place. This session was led by five Roma women leaders from different Catalan neighborhoods, also members of the LOC. These women explained how they were organizing to tackle the basic needs of Roma families, such as access to food and basic necessities.

Other women participating in the gathering joined the debate to discuss the strategies put in place to reach the most hard-to-reach Roma families and to connect them with official mechanisms of help provided by the city councils. This is observed in Carmen’s intervention, a Roma woman from Barcelona who shared the way in which, together with other neighbors, they opened a food bank, which became an essential form of relief for those most in need:

*I truly believe that Roma women can do whatever we want, because during the lockdown, many Roma women created ourselves a food bank. The work we have done is huge, and many other Roma and non-Roma organizations have joined us in our effort to guarantee basic needs for Roma families. We helped almost 350 Roma families, so definitely we can achieve whatever we propose to do*.(Carmen, participant in the gathering)

In the roundtable of experiences, some participants also shared that they saw themselves forced to go out to help Roma elders of the community, even having to face the risk of being fined by police authorities. Others also explained that they created WhatsApp groups to offer support on any issue related to COVID-19, such as sharing information about how the virus works, about public aid, or the public measures that had to be followed:

*For instance, we also created a WhatsApp group, named Gypsy Kings, where everybody could share their doubts and concerns regarding their homework, or anything during the lockdown*.(Soraya, participant in the gathering)

The dialogic context of the gathering encouraged participants to take a stand and build alliances with local schools to demand that their children (or adult) were not left behind. This was also mentioned by Irina, a young Roma woman participant in the RWSG:

*We can take the positive aspect of every situation, including this one, due to the solidarity that characterizes our culture. It has appeared even at the institutional level, for instance, when two Roma women started to sew facemasks for the entire neighborhood, and did not stop until everyone had at least one to protect themselves*.(Irina, participant in the gathering)

The experiences shared by the women revealed that, even during the lockdown, despite many of them not mastering the use of online tools, the very feature of the Roma culture of not leaving anyone alone and helping each other prevailed [34,35]. This core aspect of the Roma cultural identity made a difference when social distancing and self-isolation measures were put in place as a measure against COVID-19.

### 3.3. When the Virtual Space also Becomes Roma

Even with the digital gap pressing the Roma communities, the 19th RWSG set a milestone by demonstrating that online spaces can also be used by grassroots Roma people, and that they can become a site where the Roma community can organize and help each other. The analysis done shows that instead of engaging in the endless loop of the digital gap from which vulnerable communities are suffering (which often is the position assumed by some public authorities), the participants of the gathering and DKM organized themselves to conquer the online space and create a safe and supportive digital environment.

As revealed by the qualitative evidence gathered for this study, the above-mentioned factors are observed at two levels: first, in the examples shared by some of the women participating in the gathering about how they were already helping each other to follow and engage in all those social activities that suddenly went online, such as their children’s schooling, making health appointments, applying for job permits and formal aid related to COVID-19, and even engaging in faith events. The second was in making sure that those most vulnerable Roma women and those with poor or no digital skills would not only join but also actively participate and share their views in the activity.

### 3.4. Actions and Strategies Shared during the Gathering

As for the first level, the gathering served as a space where participants discussed and reflected on the importance of speaking up and claiming that all public services that were offered online be tailored to the needs of those most vulnerable, and those who do not have internet at home, or who have poor digital skills. Some of the women participating in the gathering explained that they engaged in conversations with local schools’ principals to defend their right to access high-quality education, expressing to them that many Roma children could not follow some of the homework assigned because their families were not able to help them in the way that was required by teachers.

However, these same women also explained that instead of waiting for the schools and other public services to do something about it, they themselves looked for a way out and put in place strategies to make sure that they did not fall behind. For instance, Celia, one of the participants, explained that she had trouble doing her homework, so she decided to ask for help from different people in the community, and ended up creating a WhatsApp study group through which they could connect and work together:

*This school year has been complicated because of COVID-19. So I created a study group with the organizations here in the neighborhood to seek help for academic support. Since I have ended the school year with good grades, I really think that we should create a similar study group like the one I had, but with Roma girls so we can support each other and keep our motivation through the academic journey*.(Celia, participant in the gathering)

A similar experience was also explained by Luz, a Roma social worker who volunteered to support Roma households without access to the internet to ensure that children could keep up with their homework during the lockdown. In the RWSG, Luz shared the work she was doing in her neighborhood with other Roma families:

*The Roma community is very supportive, whenever I go I feel we all are family. When we all come together to support our children’s education, we can make a difference, but we need the help of all of us*.(Luz, Roma social worker, participant in the gathering)

Juana, a Roma woman who joined the gathering, made a very relevant point on the way in which the Roma community is used to helping each other, emphasizing that they could take advantage of this to better cope with COVID-19. She underscored the importance of peer support, which is central in the Roma community, which they know very well how to do, and which is crucial to succeed:


*We volunteered to go to the households where the Roma families did not know how to use electronic devices and digital tools such as Zoom. This to ensure that everyone could complete their homework and keep in contact with the school (…)*
(Juana, participant in the gathering)

### 3.5. Making Sure That Everybody Who Wanted Could Join and Actively Participate

As for the second level, many grassroots Roma women who were the constituents of DKM did not have access to the internet in their households, or, as mentioned above by Juana, they did not know how to use electronic devices or online media platforms. Instead of simply accepting that these women could not join or discouraging them from participating due to technical issues, DKM set up a plan to ensure that this was not a barrier. Before the day of the event, as well as during the event, DKM offered technical help to those who asked for it to practice how the Zoom connection would work during the RWSG.

For instance, at the very beginning of the gathering, some time was allocated to explaining the overall functioning of the event and how the technical features of Zoom worked. The Roma women who moderated the gathering ensured that everyone knew how to raise their hands through the virtual platform so that participants could participate whenever they wished. Technical support was also warranted during the gathering, facilitated by Roma and non-Roma volunteers. The following moment at the beginning of the gathering reflects how this occurred:


*Moderator: “Let’s see. Let’s try it together. If we want to raise our hand, we click on “Participants”, and we then click on the icon that is hand, which means “raise hand”. Let’s try it together and see how many hands are raised… Great! Well, now we’re going to lower our hands…*



*Maria: I have a question; you touch participants and that’s it? If I do that nothing comes to me.*



*Moderator: Maria, from which type of devise are you joining? A computer or a phone?*



*Maria: from my computer*



*Moderator: and you don’t see the “raise hand” option? Well, wait… Let me ask some of our volunteers if we can call you on WhatsApp so we explain you.*
(Dialogue between the moderator explaining Zoom features and one the gathering’s participant)

Moreover, those few women who did not have an electronic device or were unwilling to join through WhatsApp and lived close to the DKM headquarters were allowed to come along and join the gathering through one of the organization’s computers. Making sure that the maximum number of people gathering at that moment was respected, eventually, nine women joined the RWSG from the DKM headquarters.

## 4. Conclusions

The results presented in this study highlight elements that nurture the resilience of the Roma community, which becomes stronger in challenging circumstances such as the present COVID-19 pandemic. Gathering 25 diverse Roma women online for the first time to search for evidence-based solutions in health and education is fully aligned with the target that the European Commission has set for 2023 regarding the participation and empowerment of Roma communities [2].

By connecting diverse Roma women in discussing issues that are of the utmost concern for the entire Roma community and the whole of society, the 19th online RWSG became and worked as a space for the expression of human agency. In so doing, the gathering counteracted the rooted prejudice that holds that the Roma do not care about science in general, or that they are against scientific advances. On the contrary, all Roma women participating in the RWSG were eager to know more about COVID-19, as well as what were the main recommendations in the field of educational research about how to make online learning attractive for all children, and how to help Roma families to support their children in their school tasks. The participants of the event communicated the lessons learned during the 19th RWSG to their extended families, which amplified the impact of the discussions and agreements made. From the individual to the community, this very Roma feature widened the scope of the debates, reaching more people.

Regarding conquering the online space, many Roma women were very active during the 19th RWSG, disseminating the main conclusions of the event on Twitter. As shown by the DKM Twitter account revised for this study (@DromKotar), many participants shared pieces of ideas discussed during the gathering. This evidence might contribute to bridging the digital gap that the Roma community has experienced, and which was exacerbated because of COVID-19 [36,37], as has happened to many other vulnerable groups worldwide [38]. Moreover, participants in the gathering highlighted the key role of public administrations in providing accessible and culturally sensitive support to Roma families, considering their particular situations when navigating the online virtual space.

This study has revealed the potential that creating a particular dialogic context—in this case, the 19th RWSG—had to connect, provide, and discuss scientific evidence with grassroots Roma women during the unique circumstances of the COVID-19 lockdown. The results presented in this study unveil the key elements of the first online RWSG, in which the Roma culture played a key role in enhancing the solidarity and resilience of this social group.

Due to the impact that the RWSG has had for more than 20 years, this action is currently being replicated in Greece, Hungry, the United Kingdom and Bulgaria through the European project RTransform (ref 621416-EPP-1-2020-1-ES-EPPKA3-IPI-SOC-IN), led and coordinated by DKM. The project is transferring the scheme and strategies for organizing a successful RWSG on the ground across Europe, opening up spaces where grassroots non-academic Roma women can engage in debates with each other, which facilitate and enhance their human agency and participation in new spaces of debate and decision-making. Thus, actions such as the RWSG are promoting the visibility of Roma women, who can be positive role models for others, while becoming community leaders and vindicating their rights as women and as part of the Roma community.

Future research could explore the impact of these gatherings in those newer settings across Europe in which they are being organized. Future research could also focus on how, by engaging in this activity, women who were once in the shadows can activate their human agency, and in some cases become community leaders, and emerge as key actors in public spaces of debate and decision-making related, for instance, to the field of education, public health, or the policy arena.

## Figures and Tables

**Table 1 ijerph-19-05524-t001:** Coding Scheme.

Thematic Dimensions	Specific Categories	Transformative Dimension	Exlusionary Dimension
Drivers for discussing the challenges derived from the lockdown with a focus on Roma children’s education	Enhancing/preventing egalitarian dialogue		
	Access to key COVID-19-related information		
	Allies built with relevant actors and stakeholders		
	Public institutions’ role		
Potential underlying elements that allowed grassroots Roma women to successfully navigate the virtual space, and which go beyond the gathering itself to the entire Roma community	Women leadership		
	Solidarity networks		
	Role played by Roma cultural traits (enhancing/preventing): extended family, intergenerational dialogue, respect of the elders, among others		

## Data Availability

The datasets used and/or analyzed during the current study are available from the corresponding author on reasonable request.

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
