# Peer review of "Connecting Roma Communities in COVID-19 Times: The First Roma Women Students’ Gathering Held Online"

_ijerph, 2022, doi:10.3390/ijerph19095524_

Round 1

Reviewer 1 Report

Thank you for the opportunity of reviewing this manuscript about inequality of certain populations (in this case the roma women) to health.

I really agree with you when you state that the solution cannot be imposed in a top-down approach, but proposed by the qualification, capacitation, empowerment and education of people.

At the same time, policies should address stigma and social integration, respecting the culture, but functioning as a way to make roma come to the occidental society and its values, instead of providing a kind of social ghettos, where traditional practices and respective powers are simply preserved through the time. The exposure of single cases like Marta (page 8) at media is crucial.

In your paper, most of actions are related to what DKM through RWSG can make (and is making) to do things happen. I think it is important to ask for what are roma population doing to make it better for their own, including efforts for integration, which we do not see most of times.

Maybe you could strengthen the text by quoting directly some of the participants in the meeting, with their own expressions.

Social networks like twitter is relevant, but we see the 3 twits with only just few likes and shares. The change will happen from the youngers and through them. Imagine, for instance, that these comments are shared in Instagram and education of roma becomes cool …

This actions may be facilitated by institutions like DKM.

Minor

Page 5: please complete the sentence “from March 15th to 21ST June 202”??

Author Response

First of all, we would like to thank both reviewers for their comments on our manuscript.

In the table below we explicate how we have addressed the comments and suggestions made by both reviewers in this first round of revision. We would like to thank for the meticulous revision done, the comments and suggestions pointed out. The overall revision process has been since its very beginning argumentative, constructive and rigorous, what indeed has contributed to improve the quality and understanding of the manuscript.

Comments and suggestions have helped us to improve the methodological analyses done and thus, to improve the overall quality of the paper. We have carefully reflected about each of them, and addressed them in the way considered most suitable.

As required, find bellow two tables including the reviewers’ comments and suggestions and our explanation of the changes made. Besides, within the manuscript, parts in which changes have been included are highlighted in yellow. Finally, a clear version of the manuscript is also included.

We believe the paper has not a higher quality and we expect that it will fit the expectations.

Best regards,

Authors

Reviewer 2 Report

Thank you for the opportunity to review this manuscript documenting an online gathering of Roma women during COVID.

General comments

I read this manuscript with interest. It was useful to read about how the documented program was valuable for participants and their communities.

Documenting the event in the academic literature is valuable as a case study of community development with a marginalised group.

The manuscript has the potential to document the views of the Roma women about survival throughout covid and the role of the online event. In addition, the methods for running the online event with a group who experienced literacy and technology access barriers is also of interest.

The manuscript would be stronger with some major edits.

Methods

The inclusion of research subjects in the research team is a strength of the research methods.

There needs to be greater detail about the methods, including the theoretical approach for the qualitative analysis of themes and the steps undertaken. Line 213 says, “Then, the two main thematic dimensions that guided the analysis procedure were identified and discussed among the researchers: a) Drivers for discussing the challenges derived from the lockdown, b) potential benefits that go beyond the RWSG itself to the entire Roma community’.

The details of the vent could be more clearly described. For example, on page 4 , line 173 the manuscript says, “The second speaker focused on Successful Educational Actions and their potential to ensure high quality education to the Roma community even during the lockdown.” It would be good to know what ‘Successful Educational Actions’ refers to.

The conclusions of the event were shared with policymakers. How did this occur? If via a written report, was this published?

Results

It would be stronger to include a greater number of quotes in the results.

It would be good to have a better link between the methods and research findings. For example, page 6 line 296 says, “The RWSG is rooted in an extensive research line devoted to promoting the best education and social opportunities for all by overcoming inequalities. Scientific evidence with social impact is at the heart of the RWSG, since it underlays the structure of the event and guides the interventions of the speakers.” The methods could describe how scientific evidence informed the structure of the event and guided the interventions.

Many of the themes documented in the results do not flow logically from the methods which described two main thematic dimensions to guide the analysis: a) Drivers for discussing the challenges derived from the lockdown, b) potential benefits that go beyond the RWSG itself to the entire Roma community.

Conclusion

The conclusion is very strong and highlights the value of the research.

Language expression

The manuscript could be edited for more consistent language. There currently seems to be a variety of styles, from academic to a more everyday style.

The manuscript seems longer than needed. The language expression could be more succinct. For example, see page 2, lines 58 to 64.

Please include the acronym in brackets at first use. See page 3, line 108.

Although use varies internationally, it is stronger in academic writing to refer to ‘Roma young people’ rather than ‘Roma youth’.

Some word use needs better clarity. For example, see ‘unconscious about’  page 3, line 130. “Unveiling this is crucial to overcome the stereotyped image of Roma women as passive and unconscious about social challenges affecting their communities, drawing pathways to strengthen the Roma community’s resilience.”

Please write numerals in full at the beginning of a sentence. For example, see page 4, line 162.

Please check for grammar. For example, page 8, line 359.

References

Please number references in order of appearance.

Some statements need references, for example, the data on page 2 line 50, documents the ‘latest international data’ but does not include a reference.

Some references are incomplete e.g. 1; 24.

Some references on health status are very old. Although they document pre-covid health status, they could be more recent e.g. 5; 6; 7

Author Response

(The authors gave the same response as above.)

Round 2

Reviewer 2 Report

This manuscript required an extensive re-write to increase clarity. Unfortunately, I don't think the edits are sufficient for publication. In particular, the research methods have not been clarified adequately. 

The research is valuable, but the manuscript is not at publication standard.

In the first round of feedback major issues were highlighted, with examples given. In this review, the authors have improved many of the specific examples that were highlighted, but they have not addressed the issues across the manuscript. 

The manuscript is very long and needs major tightening of language expression. There are many grammatical errors. 

Methods
References have been added, but the methods are very slim on details about the practical steps taken (what was done by whom). Currently, the methods describe more the philosophical approach. 

Results
The quotes from participants are a good addition; however, the quote on line 437 is too long.

Author Response

First, thank you very much for having read the revised version article for the second time and for suggesting additional revisions to improve its coherence and clarity.

Considering reviewer 2 comments, we decided to work extensively in the article, better explain the methods and how the study was designed, re-write and re-organize significantly the results, and once the draft was finished, send it for English edition.

We consider that the article is more consistent in its current shape, and that the contribution has been strengthened.

Best,